# Functional Gradients and Generalizations for Transformer In-Context Learning

## Abstract

We examine Transformer-based in-context learning for contextual data of the form $\{(x_i, y_i)\}_{i=1,N}$ and query $x_{N+1}$, where $x_i \in \mathbb{R}^d$ and $y_i \sim p(Y|f(x_i))$, with $f(x)$ a latent function. This is analyzed from the perspective of *functional* gradient descent for $f(x)$. We initially perform this analysis from the perspective of a reproducing kernel Hilbert space (RKHS), from which an alternative kernel-averaging perspective is manifested. This leads to a generalization, allowing an interpretation of softmax attention from the perspective of the Nadaraya-Watson kernel-weighted average. We show that a single attention layer may be designed to exactly implement a functional-gradient step in this setting (for RKHS latent functions), extending prior work for the special case of real-valued $Y$ and Gaussian $p(Y|f(x))$. This is also generalized for softmax attention and non-RKHS underlying $f(x)$. Though our results hold in a general setting, we focus on categorical $Y$ with $p(Y|f(x))$ modeled as a generalized linear model (corresponding specifically to softmax probability). Multi-layered extensions are developed for this case, and through extensive experimentation we demonstrate that for categorical $Y$ a single-layer model is often highly effective for in-context learning. We also demonstrate these ideas for real-world data, considering in-context classification of ImageNet data, showing the broad applicability of our theory beyond the commonly-studied settings of synthetic regression data.

## 1 Introduction

There has been significant interest recently in understanding the few-shot-learning capabilities of Transformers (von Oswald et al., 2023; Cheng et al., 2024; Ahn et al., 2023; Akyurek et al., 2022; Garg et al., 2022; Mahankali et al., 2023; Schlag et al., 2021; Zhang et al., 2023). This has been in part motivated by the impressive few-shot-learning capabilities of large language models (LLMs) (Brown et al., 2020). Much of that prior work has focused on the self-attention component in Transformers and its application to modeling real-valued functional outputs conditioned on covariates (Ahn et al., 2024; von Oswald et al., 2023; Ahn et al., 2023; Schlag et al., 2021), rather than complete Transformer layers (self-attention followed by a feedforward layer with skip connection). Further, there has been little consideration of categorical observations, which are relevant for token prediction in language modeling.

Few-shot learning is here understood as contextual learning with data $\{\mathcal{C}^{(l)}\}_{l=1,L}$, where $\mathcal{C}^{(l)} = \{(x_i^{(l)}, y_i^{(l)})\}_{i=1,N+1}$, $x_i^{(l)} \in \mathbb{R}^d$ and $y_i^{(l)}$ is an outcome (usually assumed real, but here extended to categorical). The contextual data $\{\mathcal{C}^{(l)}\}_{l=1,L}$ are used to learn a model, which employs the context $\{(x_i^{(l)}, y_i^{(l)})\}_{i=1,N}$ to constitute a prediction of $y_{N+1}^{(l)}$ (or properties thereof), for corresponding query $x_{N+1}^{(l)}$. Once such a model is learned, it is applied to new contextual data $\mathcal{C}^{(L+1)}$ for which the desired outcome $y_{N+1}^{(L+1)}$ is not available, thus it needs to be inferred.

There have been two principal approaches to this problem. One direction assumes a parametric model for the outcomes, and using $\{\mathcal{C}^{(l)}\}_{l=1,L}$ seeks to learn a good initialization point for these parameters. For new $\mathcal{C}^{(L+1)}$, the contextual data $\{(x_i^{(L+1)}, y_i^{(L+1)})\}_{i=1,N}$ are used for model-parameter refinement from the aforementioned initialization, and with these refined parameters $y_{N+1}^{(L+1)}$ is inferred for input $x_{N+1}^{(L+1)}$ (Finn et al., 2017; Nicholand et al., 2018). The alternative

approach is to use $\{\mathcal{C}^{(l)}\}_{l=1,L}$ to learn a *meta model*, that is applied to $\mathcal{C}^{(L+1)}$ with no parameter fine-tuning (Schmidhuber, 1987; Santoro et al., 2016).

The Transformer has been recognized as a few-shot learner of the latter type, *i.e.*, a meta learner. Specifically, the Transformer learns to infer a latent function $f_i^{(l)}(x)$ linked specifically to the realizations $\{f^{(l)}(x_i)\}_{i=1,N+1}$, and based on the contextual data $\{(x_i^{(l)}, y_i^{(l)})\}_{i=1,N}$ (von Oswald et al., 2023; Ahn et al., 2023; Cheng et al., 2024). It is usually assumed within the Transformer design that the latent function $f^{(l)}(x)$ resides in a particular functional class. For example, $f^{(l)}(x)$ has been assumed to be a linear model (von Oswald et al., 2023; Ahn et al., 2023; Mahankali et al., 2023), or more generally, it may reside in a reproducing kernel Hilbert space (RKHS) (Cheng et al., 2024).

Existing work studying the Transformer as an in-context learner has focused on several complementary directions. For instance, Garg et al. (2022) examined the functional classes admitted by Transformers, Muller et al. (2022) took a Bayesian analysis perspective, and others considered Transformers as functional few-shot learners (von Oswald et al., 2023; Ahn et al., 2023; Cheng et al., 2024). However, these studies have simplified the assumptions about the functional class and model architectures in favor of stronger theoretical foundations. Prominent examples include reducing the Transformer to linear attention layers (Schlag et al., 2021; von Oswald et al., 2023; Ahn et al., 2023; 2024), thus limiting the functional class to linear models (Zhang et al., 2023; von Oswald et al., 2023; Ahn et al., 2023). More recently, these ideas have been leveraged to expand the functional class to smooth functions in an RKHS with associated kernel attention (Cheng et al., 2024). These studies have provided important insights, but they have limitations in the context of some Transformer properties and applications. In this work we seek to build upon the above important contributions.

- We introduce an analytical framework for in-context learning that is applicable to settings for which the observations may be modeled as draws from an underlying model $p(Y|f^{(l)}(x))$, where $Y$ may take many forms. We connect the case of real $Y$ and Gaussian $p(Y|f^{(l)}(x))$ to prior work (Schlag et al., 2021; von Oswald et al., 2023; Ahn et al., 2023; 2024), while focusing on categorical $Y$ and softmax $p(Y|f^{(l)}(x))$.

- We show that the Transformer's forward pass can be interpreted as taking steps of *kernel averaged* functional gradient descent (Lemma 1 and Section 3). This crucially allows us to analyze the softmax attention mechanism (Proposition 1).

- We show that one attention layer may be designed to exactly perform a step along the steepest-descent direction connected to the functional gradient for categorical $Y$ and softmax model $p(Y|f(x))$ (Sections 3 and 4). This extends prior work that considered real $Y$ and Gaussian $p(Y|f(x))$ (Mahankali et al., 2023).

- We analyze function-space gradient descent dynamics when the gradient-operator is *nonlinear*, as it is for categorical and count $Y$ (and likely other cases). This highlights a potential role for a feedforward (FF) element with skip connections (Section 4).

- We examine our theoretical results in a series of experiments on both synthetic and real-world data (Section 5). Notably, we present a Transformer-based in-context classifier for the ImageNet dataset to demonstrate its efficacy and scalability.

## 2 IN-CONTEXT INFERENCE OF A LATENT FUNCTION

Assume we are given contextual data $\mathcal{C} = \{(x_i, y_i)\}_{i=1,N}$, where $x_i \in \mathbb{R}^d$ are covariates and $y_i$ is the outcome of interest. As discussed below, $y_i$ may take different forms. The goal is the development of a meta model (Schmidhuber, 1987; Santoro et al., 2016) capable of predicting a desired property of unobserved random variable $Y_{N+1}$ given context $\mathcal{C}$ and an associated query $x_{N+1}$. For example, one may desire the expectation $\mathbb{E}(Y_{N+1}|X = x_{N+1})$ (von Oswald et al., 2023; Cheng et al., 2024; Ahn et al., 2023), and in other settings the distribution $p(Y = y_{N+1}|X = x_{N+1})$. It is assumed that the probability of $Y$ conditioned on $X = x$ may be expressed as $p(Y|f(x))$, where $f(x)$ is a latent context-dependent function from a family $\mathcal{F}$.

For training the meta learner, it is assumed we are given $L$ examples of contextual data, $\{\mathcal{C}^{(l)}\}_{l=1,L}$, where $\mathcal{C}^{(l)} = \{(x_i^{(l)}, y_i^{(l)})\}_{i=1,N+1}$, and for each there is an associated latent function $f^{(l)}(x) \in \mathcal{F}$.

For notational simplicity, we assume $N$ is the same for each $\mathcal{C}^{(l)}$, but that need not be the case (this issue is discussed in detail below). When training with each $\mathcal{C}^{(l)}$, $\{(x_i^{(l)}, y_i^{(l)})\}_{i=1,N}$ are used as context, and $(x_{N+1}^{(l)}, y_{N+1}^{(l)})$ are used as the query-outcome pair for which predictions are desired. After the meta learner is trained, it is to make a prediction for query $x_{N+1}^{(L+1)}$ connected to *new* context-dependent data $\mathcal{C}^{(L+1)} = \{(x_i^{(L+1)}, y_i^{(L+1)})\}_{i=1,N}$, with new associated function $f^{(L+1)}(x) \in \mathcal{F}$, and for which $y_{N+1}^{(L+1)}$ is *unknown*. It is desired that such predictions are made without model-parameter refinement (Schmidhuber, 1987; Santoro et al., 2016); distinct from methods like MAML, which fine-tune model parameters given new contextual data (Finn et al., 2017; Nicholand et al., 2018).

For real $Y$, one may consider $p(Y|f(x)) = \mathcal{N}(f(x), \sigma^2 I)$, where the output of $f(x)$ is a vector and $I$ is the identity matrix, both of which have size defined by the dimension of $Y$. Often interest is in $\mathbb{E}(Y|X = x)$, and hence the goal is to infer $f(x)$. As discussed below, in this case $\sigma^2$ need not be inferred explicitly. For categorical $Y$, a softmax model $p(Y = y|f(x)) = \exp\left(w_y^T f(x)\right)/\sum_{c=1}^C \exp\left(w_c^T f(x)\right)$ is typically employed for $C$ categories $y \in \{1, \ldots, C\}$, and where $\{w_c\}_{c=1,C}$ represent associated embedding vectors (Vaswani et al., 2017).

For the categorical case, let $w_c$ define column $c$ of embedding matrix $W \in \mathbb{R}^{d' \times C}$. Often column $C$ is set to an all-zeros vector, and category $C$ is a reference. In one design of $W$, $d' = C - 1$ and columns $c = 1, \ldots, C-1$ are set to one-hot vectors (see Sec. C.4.1 in the Appendix of Akyurek et al. (2022), with the 1 positioned at component $c$ in column $c$. In this case the $C-1$ components of $f(x)$ are used directly within the $C$-category softmax, and logistic regression is manifested when $C = 2$. For large $C$, *e.g.*, in a language model, for which $C$ represents the number of tokens (Vaswani et al., 2017), the above design may be expensive. Hence, the second type of design for $W$ sets $d' < C - 1$ (often $d' \ll C - 1$) and for columns $c = 1, \ldots, C-1$ each $w_c \in \mathbb{R}^{d'}$ is a *learned* embedding vector.

We begin by assuming $\mathcal{F}$ corresponds to an RKHS.

**Lemma 1 (RKHS Steepest Descent.)** *Let $\kappa(\cdot, \cdot)$ denote a kernel, and let $\mathcal{F}$ be its associated reproducing kernel Hilbert space (RKHS). Given data $\{(x_i, y_i)\}_{i=1,N}$, consider the log-likelihood $L(f) = \frac{1}{N} \sum_{j=1}^N \log p(Y = y_j|f(x_j))$. Let $\{f_k\}_{k=0,1,2\ldots}$ denote the gradient descent sequence of $L(f)$ in $\mathcal{F}$ with respect to the RKHS norm, with stepsize $\alpha/N$. Then*

$$f_{k+1}(x) = f_k(x) + \frac{\alpha}{N} \sum_{j=1}^N \nabla_f \log p(Y = y_j|f_k(x_j))\kappa(x_j, x), \tag{1}$$

*where $\nabla_f \log p(Y = y_j|f_k(x_j))$ is the gradient wrt the components of $f(x)$, evaluated at $f_k(x)$.*

The lemma is proven in the Appendix, where background material on RKHS gradient descent is also reviewed. The expression $\frac{1}{N} \sum_{j=1}^N \nabla_f \log p(Y = y_j|f_k(x_j))\kappa(x_j, x)$, which is the overall gradient responsible for the update of $f(x)$ in (1), may be interpreted as a *kernel-weighted average* (Hastie et al., 2009) of the $N$ isolated gradients $\nabla_f \log p(Y = y_j|f_k(x_j))$. This interpretation of (1) will be used below to motivate a generalization.

Lemma 1 is valid for any model for the observations of the form $p(Y|f(x))$, and special cases exist in the literature for real $Y$ and Gaussian $p(Y|f(x))$. Specifically, Ahn et al. (2024); von Oswald et al. (2023); Ahn et al. (2023); Mahankali et al. (2023) consider a *linear* kernel $\kappa(x_j, x_i) = x_j^T x_i$, while Cheng et al. (2024) generalized this to any valid kernel in RKHS. Here we generalize the forms of $Y$ that may be considered.

In particular, for real $Y$ with Gaussian $p(Y|f(x))$ where $f(x)$ is the mean and the covariance is $\sigma^2 I$, and for categorical $Y$ with softmax $p(Y|f(x))$, we have

$$\text{Real } Y \quad : \quad \nabla_f \log p(Y = y_j|X = x_j) = y_j - f_k(x_j), \tag{2}$$

$$\text{Categorical } Y \quad : \quad \nabla_f \log p(Y = y_j|X = x_j) = w_{y_j} - \mathbb{E}(w_c)_{|_{f_k(x_j)}}, \tag{3}$$

where $\mathbb{E}(w_c)_{|_{f_k(x_j)}}$ is the expectation over the embedding vectors, using softmax probability based on $f_k(x_j)$. Equations (2), (3) and $\mathbb{E}(w_c)_{|_{f_k(x_j)}}$ are derived in the Appendix.

From (2), note that for real $Y$ the residual (error) $y_j - f_k(x_j)$ informs the update direction in the gradient for refinement of $f_k(x)$, with the degree to which sample $j$ informs the update for $f_k(x)$ dictated by closeness in covariate space, quantified via the kernel $\kappa(x_j, x_i)$. For categorical $Y$, $w_{y_j} - \mathbb{E}(w_c)|_{f_k(x_j)}$ plays an analogous role: as $f_k(x_j)$ is more peaked (probable) for category $y_j$, the expectation will be close to $w_{y_j}$, so $w_{y_j} - \mathbb{E}(w_c)|_{f_k(x_j)}$ reflects a form of model error wrt the contextual data, to which the model adapts at inference.

While below we will focus on categorical $Y$ and make connections to real $Y$ for which there is much prior work, in the Appendix we also show that for count-valued $Y$ modeled as $\text{Poisson}[\exp(f(x)]$, the gradients similarly form a model error of the form $y_j - \mathbb{E}(y_j)|_{f_k(x_j)}$, and learning consists of pushing $f_k(x_j)$ to make the expected counts consistent with observations.

Lemma 1 assumed that each component of vector function $f(x)$ is within the RKHS family with kernel $\kappa(x_j, x_i)$. However, having arrived at this form, we note that the overall gradient for $f(x)$ is effectively a kernel-weighted average over isolated gradients $\{\nabla_f \log p(Y = y_j | f(x_j))\}_{j=1,N}$ (Hastie et al., 2009). We now seek to employ this insight, and move beyond the assumption that $f(x)$ is in a RKHS.

**Proposition 1 (Nadaraya-Watson Averaged Gradient Descent)** *Consider data* $\{(x_i, y_i)\}_{i=1,N}$ *and assume that the* $y_i$ *are drawn from* $p(Y|f(x_i))$ *for an underlying function* $f(x)$. *The corresponding Nadaraya-Watson average gradient, for a nonnegative kernel* $K_\lambda$ *with parameter* $\lambda$ *(which here need not be positive semi-definite) is*

$$f_{k+1}^{NW}(x) = f_k^{NW}(x) + \alpha \sum_{j=1}^N \nabla_f \log p(Y = y_j | f_k^{NW}(x_j)) \left[ \frac{K_\lambda(x_j, x)}{\sum_{j'=1}^N K_\lambda(x_{j'}, x)} \right]. \quad (4)$$

This is a direct application of the Nadaraya–Watson kernel-weighted average (Nadaraya, 1964; Watson, 1964; Hastie et al., 2009) to the sample-dependent gradients $\{\nabla_f \log p(Y = y_j | f(x_j))\}_{j=1,N}$, without assumption on the form of $f(x)$, yielding a generalization of (1). An important special case is $K_\lambda(x_j, x) = \exp(\lambda x^T x_j)$ with $\lambda > 0$, and for which $K_\lambda(x_j, x)/\sum_{j'=1}^N K_\lambda(x_{j'}, x)$ corresponds to softmax attention.

Note that in (4) there is *no* $1/N$, as the average is manifested via the denominator in $K_\lambda(x_j, x)/\sum_{j'=1}^N K_\lambda(x_{j'}, x)$, which is distinct from (1). This will be revisited in the experiments.

## 3 META-LEARNING MODEL CONSTRUCTION

Based on the understanding of functional gradients from above, we now examine the form of our desired meta learner. We focus on multi-layered models, in which the importance of skip connections (He et al., 2015; Drozdzal et al., 2016) becomes evident. Additionally, in the remainder of the paper we introduce the notation $f_i = f(x_i)$ (and related usages will become evident below). We use this notation to simplify the equations, and hope that this will not cause confusion with the RKHS notation used earlier.

Let $\{h_i^{(\ell)}\}_{i=1,N}$ represent the hidden inputs to layer $\ell + 1$, with $\{h_i^{(\ell)}\}_{i=1,N}$ output from layer $\ell$ (the initial inputs correspond to $\ell = 0$). The set $\{h_i^{(\ell)}\}_{j=1,N}$ is *collectively* analyzed at layer $\ell + 1$, and at the output of layer $\ell + 1$ are emitted $\{h_i^{(\ell+1)}\}_{i=1,N}$. We will show in the next section that $h_i^{(\ell)}$ has three sets of components corresponding to the covariates, function evaluations and gradient information, but for now it suffices to assume that a *subset* of the components of $h_i^{(\ell)}$ correspond to the model of $f_i$ after $\ell$ layers, denoted $f_i^{(\ell)}$, while other components of $h_i^{(\ell)}$ play a role in the computation of $f_i^{(\ell+1)}$.

With skip connections, the cumulative representation of $f_i^{(\ell+1)}$ at the output of layer $\ell+1$ is $f_i^{(\ell+1)} = f_i^{(\ell)} + g_\theta(i; \{h_j^{(\ell)}\}_{j=1,N})$, where $g_\theta(i; \{h_j^{(\ell)}\}_{j=1,N})$ denotes the output of the model at this layer for representing $f_i$ (other parts of $h_i^{(\ell+1)}$ may also be updated, in addition to $f_i^{(\ell+1)}$, but we focus on the latter as it is involved in the likelihood fit to the data). The model parameters are represented by

$\theta$, and each layer will in general have different parameters (but the same model architecture). We wish to examine desired properties of the meta learner composed of the models $g_\theta(i; \{h_j^{(\ell)}\}_{j=1,N})$, for performing effective inference of the latent $f_i$ given contextual data.

The log-likelihood of the data $\{(x_i, y_i)\}_{i=1,N}$ based on the output at layer $\ell + 1$ is $\sum_{i=1}^{N} \log p(Y = y_i | f_i^{(\ell+1)})$. From a Taylor series expansion we have

$$\sum_{i=1}^{N} \log p(Y = y_i | f_i^{(\ell)} + g_\theta(i; \{h_j^{(\ell)}\}_{j=1,N})) \tag{5}$$

$$= \sum_{i=1}^{N} \log p(Y = y_i | f_i^{(\ell)}) + \sum_{i=1}^{N} \left[\nabla_{f_i} \sum_{j=1}^{N} \log p(Y = y_j | f_j^{(\ell)})\right]^T g_\theta(i; \{h_{j'}^{(\ell)}\}_{j'=1,N}) + \delta,$$

where $\nabla_{f_i} \sum_{j=1}^{N} \log p(Y = y_j | f_j^{(\ell)})$ is defined by the functional gradients from the previous sections: ($i$) from the RKHS perspective, it corresponds to $\sum_{j=1}^{N} \nabla_f \log p(Y = y_j | f_j^{(\ell)}) \kappa(x_j, x_i)$, and ($ii$) from the Nadaraya-Watson perspective it corresponds to $\sum_{j=1}^{N} \nabla_f \log p(Y = y_j | f_j^{(\ell)}) K_\lambda(x_j, x_i) / \sum_{j'=1}^{N} K_\lambda(x_{j'}, x_i)$. In (5), $\delta$ represents higher-order terms, related to second and higher-order gradients of $\sum_{j=1}^{N} \log p(Y = y_j | f_j^{(\ell)})$.

To first order, if we analyze each layer one-by-one as we proceed through inference of the latent function (stepwise increase of $\ell$), (5) indicates that each $g_\theta(i; \{h_{j'}^{(\ell)}\}_{j'=1,N})$ should approximate a functional gradient step, *i.e.*, $g_\theta(i; \{h_{j'}^{(\ell)}\}_{j'=1,N}) \approx \alpha \nabla_{f_i} \sum_{j=1}^{N} \log p(Y = y_j | f_j^{(\ell)})$, for some scalar $\alpha > 0$ (which typically will depend on layer $\ell$). Hence, to first order, a series of layers may be viewed as characteristic of multiple steps of functional gradient descent, assuming $g_\theta(i; \{h_j^{(\ell)}\}_{j=1,N})$ is designed to have a large inner product (alignment) with $\nabla_{f_i} \sum_{j=1}^{N} \log p(Y = y_j | f_j^{(\ell)})$.

As developed in Cheng et al. (2024); von Oswald et al. (2023); Ahn et al. (2023); Mahankali et al. (2023), and summarized in the Appendix, for real $Y$ and Gaussian $p(Y|f(x))$, attention networks may be designed to implement each $g_\theta(i; \{h_{j'}^{(\ell)}\}_{j'=1,N})$ exactly as a functional gradient step. In such a setup, each attention layer of a Transformer implements a step of functional gradient descent on the model's forward pass. However, when considering multiple layers, better performance can be achieved by accounting for higher-order terms connected to $\delta$ in (5), yielding what has been termed GD++ (von Oswald et al., 2023) and more generally, *preconditioned* functional gradient descent (von Oswald et al., 2023; Ahn et al., 2023), also implemented in the Transformer's forward pass thus constituting the meta learner.

For categorical $Y$ and softmax $p(Y|f(x))$, we show below that a one-layer model $g_\theta(i; \{h_j^{(0)}\}_{j=1,N})$ can be represented by an attention layer to exactly implement one functional gradient step (which is also true for real $Y$ and Gaussian $p(Y|f(x))$ (von Oswald et al., 2023; Cheng et al., 2024; Ahn et al., 2023; Akyurek et al., 2022; Garg et al., 2022; Mahankali et al., 2023; Schlag et al., 2021; Zhang et al., 2023)). For single-layer models and general $Y$ and $p(Y|f(x))$, the initial direction of $g_\theta(i; \{h_j^{(0)}\}_{j=1,N})$ along which $-\nabla_{f_i} \sum_{j=1}^{N} \log p(Y = y_j | f_j^{(0)})$ decreases fastest corresponds to the functional gradient. In this sense, that *direction* may be viewed as optimal. However, the appropriate size of this gradient step depends on the problem of interest, and it is *learned* based on the contextual training data $\{\mathcal{C}^{(l)}\}_{l=1,L}$. We demonstrate in Section 5 that a single layer of attention (single functional-gradient step) typically yields effective results for categorical $Y$ and softmax $p(Y|f(x))$, with minimal improvement with further layers.

## 4 TRANSFORMER DESIGN FOR CATEGORICAL OBSERVATIONS

**Single-layer of Attention, Exact Single Functional Gradient step** A single layer of attention may be designed to exactly implement one functional-gradient step, under the RKHS assumption (connected to Lemma 1) or under the Nadaraya-Watson representation (connected to Proposition 1), for real or categorical $Y$. Real $Y$ has been considered in Cheng et al. (2024) for RKHS functions

$f(x)$, and it is readily extended via Nadaraya-Watson to non-RKHS attention. We summarize this setup in the Appendix. Here we focus on the case of categorical $Y$

Consider the encoding $h_i^{(0)} = (x_i, 0_{d'}, w_{y_i} - \frac{1}{C}\sum_{c=1}^{C} w_c)^T$ for $i = 1, \ldots, N$, and $h_{N+1}^{(0)} = (x_i, 0_{2d'})^T$. This form of encoding is similar to that considered previously for real $Y$ (see von Oswald et al. (2023); Cheng et al. (2024) and the Appendix). Specifically, each $h_i^{(0)}$ includes the covariates $x_i$ as well as the initial corresponding functional gradient, the latter represented by $w_{y_i} - \frac{1}{C}\sum_{c=1}^{C} w_c$. Each $f_i$ is initialized as a $d'$-dimensional all-zeros vector (also done in von Oswald et al. (2023) for real $Y$ of vector dimension $d'$), represented by $0_{d'}$ in $h_i^{(0)}$.

The key/query decomposition is the same as for real $Y$ (von Oswald et al., 2023; Cheng et al., 2024), *i.e.*, positions $i = 1, \ldots, N$ are used as keys, and all positions $i = 1, \ldots, N+1$ are used as queries. Using widely used notation from the Transformer literature (Vaswani et al., 2017; Brown et al., 2020; von Oswald et al., 2023; Cheng et al., 2024; Ahn et al., 2023; Akyurek et al., 2022; Garg et al., 2022; Mahankali et al., 2023; Schlag et al., 2021; Zhang et al., 2023), for a single-layer model, consider matrices $W_K = W_Q$ designed such that $W_K h_i^{(0)} = (x_i, 0_{d'+d})^T$, and matrix $W_V$ is designed such that $W_V h_i^{(0)} = (0_d, \frac{\alpha}{N}[w_{y_i} - \frac{1}{C}\sum_{c=1}^{C} w_c], 0_{d'})^T$. Finally, consider the matrix $P = I_{d+2d'}$, where $I_{d+2d'}$ is the corresponding identity matrix. The explicit forms of $W_K, W_Q, W_V$ and $P$ are provided in the Appendix.

For query $W_Q h_i^{(0)}$, we implement the attention $P\sum_{j=1}^{N} W_V h_j^{(0)} \kappa(W_Q h_i^{(0)}, W_K h_j^{(0)})$, as in Cheng et al. (2024). This is added to $h_i^{(0)}$ via the skip connection, yielding $h_i^{(1)} = (x_i, f_i^{(1)}, w_{y_i} - \frac{1}{C}\sum_{c=1}^{C} w_c)^T$, with $f_i^{(1)} = \frac{\alpha}{N}\sum_{j=1}^{N}(w_{y_j} - \frac{1}{C}\sum_{c=1}^{C} w_c)\kappa(x_i, x_j)$ corresponding to the functional gradient (with a similar representation from the perspective of the Nadaraya–Watson kernel-weighted average and softmax attention). At the output of this attention layer we specify a $d' \times (d + 2d')$ matrix denoted as $Z$ (and defined explicitly in the Appendix) acting at the output for position $N+1$ (a similar construct at the output has been used for real $Y$ (von Oswald et al., 2023)), selecting the desired latent vector from $h_{N+1}^{(1)}$ as $f_{N+1}^{(1)} = Z h_{N+1}^{(1)}$. Vector $f_{N+1}$ is sent to the softmax layer for the category probabilities. Such an output matrix $Z$ applied at the output in this manner for all models considered here.

**Multi-Layered GD-motivated Transformer, Linear Approximation** While for real $Y$ the $f_j$ may be updated as a *sum* of differential updates (see (8) in the Appendix), there is a nonlinear dependence of $\mathbb{E}(w_c)_{|_{f_i^{(\ell)}}}$ on $f_i^{(\ell)}$ (via the softmax probability over categories). Our simplest approach for addressing this complexity considers an additive representation of the needed expectation, analogous to real $Y$: $\mathbb{E}(w_c)_{|_{f_i^{(\ell)}}} = \sum_{\ell'=0}^{\ell-1} \Delta_{\mathbb{E}_i}^{(\ell')}$, where $\Delta_{\mathbb{E}_i}^{(0)} = \frac{1}{C}\sum_{c=1}^{C} w_c$. Further, we model $\Delta_{\mathbb{E}_i}^{(\ell)}$ for $\ell > 0$ via the linear approximation as $\Delta_{\mathbb{E}_i}^{(\ell)} = B^{(\ell)}\Delta_{f_i}^{(\ell)}$, where $\Delta_{f_i}^{(\ell)}$ is the differential update to the underlying function manifested by the attention layer: $\Delta_{f_i}^{(\ell')} = \frac{\alpha}{N}\sum_{j=1}^{N}[w_{y_i} - \sum_{\ell'=0}^{\ell-1}\Delta_{\mathbb{E}_i}^{(\ell')}]\kappa(x_j, x_i)$. The term $\Delta_{f_i}^{(\ell)}$ is updated using attention as in the previous section, via the same $W_Q, W_K$ and $W_V$ considered there (now used at all layers). In this approximation, there is an additional $d' \times d'$ matrix $B^{(\ell)}$ to be learned at each layer.

While the matrices $W_K, W_Q, W_V$ are unchanged from above, $P$ is modified slightly, to place $\Delta_{f_i}^{(\ell')}$ in a position to update the model of $f_i$ (with the additive skip connection), and the term $-B^{(\ell)}$ is multiplied with $\Delta_{f_i}^{(\ell)}$ to yield the aforementioned approximation to the differential expectation, which is added to the position of the expectation in the latent vector (again via skip connection). See the Appendix for the detailed form of $P^{(\ell)}$, and all attention matrices, omitted here for brevity.

**Multi-Layered GD-motivated Transformer, Nonlinear Approximation** We now consider a *nonlinear* Transformer *block*, with each block consisting of an attention layer, followed by identical feedforward networks that operate individually on each of the vectors output from the attention layer. This model is motivated by the goal of addressing the *nonlinear* dependence of $f_i^{(\ell)}$ on $\mathbb{E}(w_c)_{|_{f_i^{(\ell)}}}$ without a linearization, and it yields a model closely linked with the original Transformer design (Vaswani et al., 2017; Brown et al., 2020).

We reuse the attention mechanism from above ($W_Q$, $W_K$, $W_V$ and $P$ are unchanged), and from that analysis the output from the first attention layer is $h_i^{(1)} = (x_i, f_i^{(1)}, w_{y_i} - \Delta_{\mathbb{E}_i}^{(0)})^T$, where we use the notation $\Delta_{\mathbb{E}_i}^{(\ell)}$ from the previous subsection (which for $\ell = 0$ corresponds to the average over category embedding vectors).

We focus here on the first Transformer block, but the same construction is used at each layer. With the feedforward element and an associated skip connection, we wish to add $(0_{d+d'}, -\Delta_{\mathbb{E}_i}^{(1)})^T$ to the output of the attention. This same type of differential update to the expectation was used above under a linear approximation, and here we now consider a *nonlinear* representation.

A multi-layered perceptron (MLP) is used for this nonlinear element, with a skip connection, and the input is $h_i^{(1)} = (x_i, f_i^{(1)}, w_{y_i} - \Delta_{\mathbb{E}_i}^{(0)})^T$. In our GD-motivated construction, we do not anticipate dependence on $x_i$, so these are "zeroed-out" within the feedforward (FF) design (within the *GD* model in which we impose structure), and a $d'$ dimensional output is placed in the desired location of $-\Delta_{\mathbb{E}_i}^{(1)}$, which is made convenient via the skip connection: $h_i^{(1)} \leftarrow h_i^{(1)} + (0_{d+d'}, g_{\phi^{(1)}}(h_i^{(1)}))^T$. The updated $h_i^{(1)}$ is the total output from the first Transformer block. This attention mechanism followed by FF element is as employed in Transformers (Vaswani et al., 2017), from which it is also motivated. See the Appendix for more details.

## 5 EXPERIMENTS

Following von Oswald et al. (2023); Cheng et al. (2024); Ahn et al. (2023), we consider two training paradigms. The first is termed *GD* for (functional) gradient descent, in which all Transformer matrices are set as defined in the previous section (explicit details in the Appendix). For GD, when the embedding vectors $\{w_c\}_{c=1,C}$ are to be learned (*not* set as one-hot (Akyurek et al., 2022)), they are also learned with GD training. The second model design is termed *Trained TF* (fully trained Transformer), for which the model construction is as in the previous section, but *all* parameters are learned, without constraints. Under the linearization of Section 4, the Trained TF counterpart corresponds to attention alone at each layer, with full flexibility in learning the parameters. The Trained TF version of the model with feedforward (FF) elements consists of a general (learned) attention layer followed by FF elements (with skip connections for both), like in the original Transformer (Vaswani et al., 2017). For both the GD and Trained TF setups, when present, the feedforward element has $10d'$ hidden units and GELU activation.

For GD and Trained TF, the training seeks to minimize the cross-entropy loss of predicting $y_{N+1}^{(l)}$ for query $x_{N+1}^{(l)}$ based on context $\{(x_i^{(l)}, y_i^{(l)})\}_{i=1,N}$, for contextual training sets $l = 1, \ldots, L$. The softmax representation for $p(Y|f(x))$ is used explicitly in this cross-entropy loss. When the embedding vectors for the categories are learned, their role in the softmax probability over categories is accounted for, as well as the encoding at the embedding vectors at the input (as in language models (Vaswani et al., 2017)). The performance of the model is assessed based on average prediction accuracy of $y_{N+1}^{(L+m)}$ for query $x_{N+1}^{(L+m)}$, and also log-likelihood fit to $y_{N+1}^{(L+m)}$ conditioned on $x_{N+1}$, for $M$ distinct test sets, $m = 1, \ldots, M$ (average performance over these is shown). Below we discuss difficulties observed when training the Trained TF; because of that, for Trained TF training we also considered a separate validation dataset of size $L$, and employed early stopping. Computations were performed on a V100 GPU, and training was done with Adam (Kingma & Ba, 2015).

There has been significant recent attention directed to the challenges of learning Transformer parameters, for language models (Liu et al., 2023). Training challenges have also been examined for in-context learning applications like studied here (Ahn et al., 2024) (but there for real $Y$, linear $f(x)$ and Gaussian $p(Y|f(x))$). For Trained TF (but not GD) we observed similar challenges for categorical $Y$ and softmax $p(Y|f(x))$, and used methods related to those in Liu et al. (2023) to address them. These issues are discussed in detail below.

We first consider simulated data. Specifically, a softmax generative model $p(Y = c|f(x)) = \exp[w_c^T f(x)]/\sum_{c'=1}^{C} \exp[w_{c'}^T f(x)]$, for $C = 25$ and embedding dimension $d' = 5$. For data synthesis, $w_c \in \mathbb{R}^{d'}$ were generated randomly (and then fixed to generate data), with each component drawn i.i.d. from $\mathcal{N}(0, 1)$. The function $f^{(l)}(x)$ is different for each contextual set $l$. Specifically,

for each context $l$, 5 categories are selected uniformly at random from the dictionary of $C = 25$ categories. Let $c^{(l)}(1), \ldots, c^{(l)}(5)$ denote these categories for context $l$. We further randomly generate 5 respective "anchor positions," $\tilde{x}(1), \ldots, \tilde{x}(5)$, each drawn i.i.d. from $\mathcal{N}(0_d, I_d)$. The function for context $l$ is represented as $f^{(l)}(x) = \lambda \sum_{m=1}^{5} w_{c(m)} \kappa_{RBF}[x - \tilde{x}(m); \sigma_m]$, where the RBF kernel parameter $\sigma_m$ is selected such that $\kappa_{RBF}[x - \tilde{x}(m); \sigma_m] = \exp(-\sigma_m^2 \|x - \tilde{x}_m\|_2)$ equals 0.1 at the center of the other kernel to which it is closest (in a Euclidean distance sense). We set $\lambda = 10$ (selected so as to have category $c(m)$ clearly most probable in the region of $\tilde{x}(m)$), and $d = 10$. This model is designed such that category $c(m)$ has relatively high (and context-dependent) priority in the vicinity of center $\tilde{x}(m)$, with $\sigma_m$ defining the region of relevance for that category.

After $f^{(l)}(x)$ is designed for context $l$, $N + 1$ covariates are drawn $x_i^{(l)} \sim \mathcal{N}(0_d, I_d)$, and then $\{y_i^{(l)}\}_{i=1,N+1}$ are drawn from the underlying softmax model (at test $y_{N+1}^{(l)}$ is the true category, and the Transformer yields a probability of all categories $c \in \{1, \ldots, N\}$ given $x_{N+1}^{(l)}$). When training $N = 125$, and in all of these simulated experiments the test contextual data also has $N = 125$, except for an experiment that examines the situation for which the training $N$ is different from the test contextual size. We considered $L = 2048$ training contextual sets, and tested performance for $M = 2048$. Any contextual set $\mathcal{C}^{(l)}$ will likely only see a small subset of the total $C = 25$ categories, analogous to how in language contextual data only sees a subset of all tokens.

When training the Transformer, for this experiment the embedding vectors $\{w_c\}_{c=1,C}$ are learned, with $d' = 5$ (although for these simulated data similar results were found for other settings, such as $d' = 10$). Finally, in all cases for which model parameters are initialized at random, results from five random parameter initializations are shown, to indicate variability in the training process.

Figure 1 considers results for a *one-layer* Transformer, presenting GD and Trained TF results; for both of these, *all* parameters are initialized at random (drawn from a zero-mean Gaussian). Note that there is a significant difference in the number of parameters that need be learned for GD and the Trained TF: for GD, the Transformer need only learn the attention kernel parameter, the learning rate, and the embedding matrix $W$, while for Trained TF the embedding matrix $W$ and all the Transformer matrices $W_Q, W_K, W_V, P$ and $Z$ are learned. As observed in Figure 1,

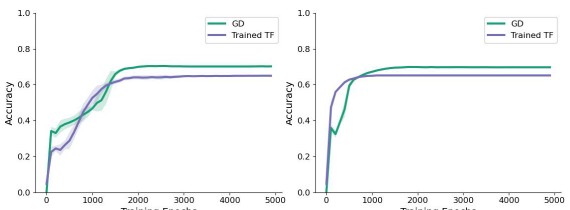

Figure 1: Average accuracy (on test data) of a one-layer Transformer with $C = 25$ categories, trained via GD and Trained TF, in both cases from a random parameter initialization. Left: softmax attention, right: RBF attention.

the Trained TF model learns relatively well, but it does not achieve the performance of GD. The GD setting constitutes a subset of all possible parameter settings for Trained TF; the fact that GD results are notably better than those of the Trained TF reflects that learning of Trained TF starting from random initialization has significant difficulties. From Section 3, the efficacy of the GD is expected, which may be viewed as an upper bound on what Trained TF could achieve.

Following Liu et al. (2023), it is recognized that proper initialization of the Transformer parameters is critical (when all parameters are free to learn, as in Trained TF). In Figure 2 we next consider training the Trained TF with parameters initialized as those from GD. We show the full training trajectory for Trained TF (starting from the final GD parameters) and for GD, and see that the Trained TF performance barely moves from the final GD performance. Separately, Figure 2 (right) compares GD performance for a linear kernel attention, showing that it performs significantly worse than nonlinear RBF and softmax attention (as expected for the highly nonlinear $f^{(l)}(x)$ used in data generation).

We next examine a two-layer model, considering each of the methods discussed in Section 4 for addressing the nonlinear expectation $\mathbb{E}(w_c)_{f_i^{(\ell)}}$. Figure 3 (left) indicates that the GD results converge to almost the same predictive accuracy, for one and two layers. For the two-layer Trained TF, it was even more important to initialize the parameters well, doing so from the learned GD parameters. In Figure 3 we show two-layer results using the linear (attention-layer-only), and the setup that used the additional feedforward element. Both approaches for multi-layered models give similar results, and do not vary significantly from the performance of the GD model from which they were initialized.

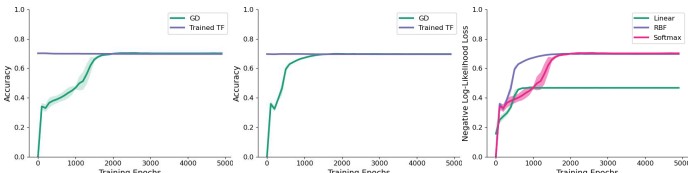

Figure 2: The left two figures consider the same case as the in Figure 1, but here the Trained TF is initialized with the GD model parameters. Right: GD performance for linear, RBF and softmax attention.

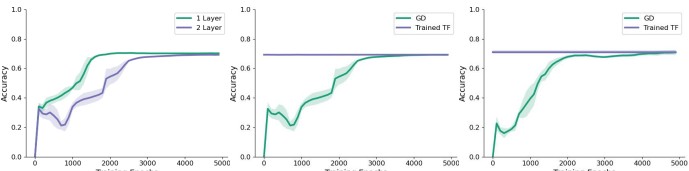

Figure 3: Left: Average GD performance as a function of training epochs, for one and two Transformer layers (for 2 layers, we show results with the FF element). The right two figures are GD and Trained TF performance for two layers, with Trained TF initialized with the GD parameters. Center: two-layer model based on linear approximation; Right: two-layer model with feedforward element. All results for test data.

Two observations are made from these results: $(i)$ the linear method of approximating $\mathbb{E}(w_c)_{|_{f_i^{(\ell)}}}$ (attention alone) yields results comparable to those that include a feedforward element and associated skip connection, and $(ii)$ none of the two-layer results are significantly different from those of the one-layer model. This phenomenon was observed in all of our experiments. We attribute $(ii)$ to the fact that the attention layer and skip connection manifest an effective functional gradient step (Section 3) with one layer. Another factor for $(ii)$ concerns the characteristics of categorical $Y$ and softmax $p(Y|f(x))$. Our predictions use the most-probable category as our prediction for query $x_{N+1}$. Even if the *overall* softmax function $p(Y|f^{(\ell)}(x_{N+1}))$ changes with increasing layers $\ell$, if the most-probable category is manifested after $\ell = 1$ layers, no change in predictive performance will be observed for more layers.

Thus far we have observed little difference between performance with an RBF or a softmax kernel. There is therefore a question of whether softmax attention offers advantages. As discussed in comparing (1) and (4), the inherent normalization in the softmax avoids the $1/N$ scaling in (1); the $1/N$ is needed to achieve a context-dependent *average* negative-log-likelihood fit.

In Figure 4 (left) we compare performance of a GD-based single-layer Transformer trained with $N = 125$, and applied to text data with $N$ varying from 25 to 300. For the linear and RBF attention kernels, we show results for which the $1/N$ term within the Transformer is left unchanged from training, and also with the software adjusted appropriately for each test $N$ (all learned parameters the same, with just an adjustment for $N$ at test). Note that the kernel attention models perform relatively stably with varying $N$ (*if* there is a rescaling for each test $N$). While this adjustment could be done in practice, it adds a complexity. The Transformer based on softmax attention yields relatively stable results for all test $N$, with no parameter adjustment at test time, and softmax-attention performance is better than all other kernels, with or without rescaling the $N$.

Our final results consider few-shot learning for classifying image data. Using the ImageNet dataset (Russakovsky et al., 2014), we select 900 classes for Transformer training, and a separate 100 classes for testing. For each contextual set $\mathcal{C}^{(l)}$, 5 distinct classes are selected uniformly at random, and for each such class 10 specific images are selected at random, and therefore $N = 50$ (image $N + 1$ is selected at random from the 5 class types considered in the context data). When training, $L = 2048$ and test performance is averaged over $M = 2048$ contextual sets. The covariates $x_i$ (with $d = 512$) correspond to features from the VGG network (Simonyan & Zisserman, 2015). We set $d' = 4$ and achieved best results (reported here) with learned $\{w_c\}_{c=1,C}$.

While this is viewed as an interesting large-scale test, we note that it does not conform to the assumptions of most prior in-context learning with Transformers (von Oswald et al., 2023; Cheng et al., 2024; Mahankali et al., 2023). In most prior work (and in the above experiment with synthetic data) it is assumed that the covariates are drawn from the same distribution for all contextual data blocks. Since here each contextual block will be associated with only 5 image classes, and in general

with a different 5 images per contextual block, the covariates (VGG features) will in general be sampled from a different distribution for each contextual block. Even more importantly, when testing the learned Transformer, it will be applied to contextual sets from entirely different image classes, thereby being drawn from a different portion of VGG feature space. We compare Transformer results to "linear probing," in which a linear model is applied to the VGG features, the outcome from which is then sent into the softmax for classification. Importantly, with linear probing a model *must be learned anew* for each test contextual block, to be contrasted with the Transformer, for which no fine-tuning of parameters are done after training. The linear probing results should be viewed as giving a sense of the quality of the Transformer results (providing a likely unachievable upper bound). While an interesting comparison, it is not an entirely "fair" comparison, as linear probing is *not* learning a single meta model (as the Transformer is).

In the right part of Figure 4 we present top-1 accuracy and the negative log-likelihood for 1 layer of attention, trained with GD, softmax attention, with test-set results shown as a function of training iteration (using parameters, at test, corresponding to that training step). While the Transformer does not achieve the performance of linear probing, it is close, and is achieved with no post-training fine-tuning and with the aforementioned mismatch in covariate statistics. In the Appendix we also show similar results for the same Transformer, but tested on $N = 15$.

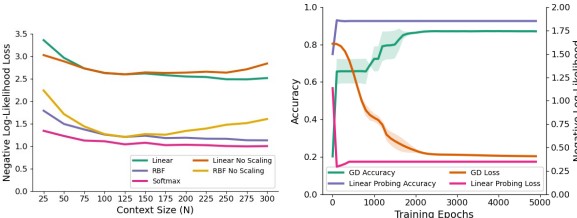

Figure 4: Left: GD-based results for a one-layer Transformer, on the synthetic data. The context size for training is $N = 125$, and we test performance for context sizes of $N = 25$ through $N = 300$. Right: Results for ImageNet dataset, with a single-layer GD Transformer with softmax attention trained with $N = 50$. For GD training, 5 random parameter initializations were considered, and variation is depicted. Results are shown as a function of the training iterations (tested on the held-out set). Also shown are results for linear probing, where in that case a new model is trained for each contextual block. Results are shown for top-1 predictive accuracy and the negative log-likelihood loss for $y_{N+1}$.

## 6 CONCLUSIONS

The above analysis demonstrates the effectiveness of Transformers for performing in-context learning for data $Y$ drawn from a model of the form $p(Y|f(x))$ for context-dependent and generally nonlinear $f(x)$. The analysis was motivated by gaining understanding of Transformer mechanisms, but we have also found that the simplified representations uncovered may serve as good initializations of parameters for less constrained Transformer representations. There are several issues that deserve further study. For inference with categorical $Y$, we observed that a single-layer model is often sufficient. For multiple layers, to model the nonlinear $\mathbb{E}(w_c)_{|f_i}$, we considered attention layers alone (linear approximation to expectation), and Transformer blocks composed of an attention layer followed by feedforward elements; the latter affording significant modeling flexibility. Nevertheless, we found a single-layer model sufficient, providing possible explanations.

We also discussed the difficulty of learning the Trained TF Transformer parameters from a random initialization, consistent with the analysis in Liu et al. (2023). Given the difficulty of training Transformers, one may wonder if the lack of observed improvement in performance with more layers may be caused by training difficulties. While this must be explored further, we note that the GD-based model, with far fewer parameters to learn than Trained TF, also did not manifest significant improvement with more layers (even when we invoked the modeling sophistication of the FF element).

The fact that the FF element did not yield significant improvements here should not be taken to imply that it is unimportant in other applications. An aspect of the in-context learning considered here is the presence of covariates $x_i$ (like done previously for real $Y$ (von Oswald et al., 2023; Cheng et al., 2024; Ahn et al., 2023)). While we encoded our input-data categories with embedding vectors (like in language models), in language models there are no covariates $x_i$ (Vaswani et al., 2017; Brown et al., 2020). One conjecture is that language models may predict the next token using inference like that considered here (with that inference performed near the output layers of a large language model). However, if this conjecture is true, it is possible that many of the other (numerous) layers of the Transformer nearer to the input layer may be performing context-dependent feature generation (analogous to inferring features that play a role *like* $x_i$). Much more work is needed to explore this and other issues uncovered in this research.

ETHICS STATEMENT

The authors have undertaken to perform the research presented here with the highest level of ethics, in accordance with the ICLR Code of Ethics. In the paper, it is believed that there are no matters that arise ethical concerns: there have been no studies that involve human subjects, we have used no proprietary datasets, we do not believe we have uncovered any harmful insights, methodologies and applications, there are no potential conflicts of interest and sponsorship, discrimination/bias/fairness concerns, privacy and security issues, legal compliance, and research integrity issues.

REPRODUCIBILITY

The authors have used no proprietary datasets. All data considered here is in the public domain (*e.g.*, the ImageNet dataset), and we have explained in detail how all of our experiments have been done, so that they may be reproduced. Further, we intend to make public all software we have developed to implement our experiments. We have included an Appendix, wherein all of the details of the theoretical developments and models are provided, to enhance understanding and reproducibility.

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

## A  APPENDIX

### A.1  PROOF OF LEMMA 1

In the following, we also include background information on functional gradients and gradient descent for functions in an RKHS.

Assume that $\mathcal{F}$ corresponds to a reproducing kernel space (RKHS) (Scholkopf & Smola, 2002; Cheng et al., 2024), which includes linear models (Ahn et al., 2024; von Oswald et al., 2023; Ahn et al., 2023; Mahankali et al., 2023) as a special case. For input $x \in \mathbb{R}^d$, we first consider function $f(x) \in \mathcal{F}$ with a scalar output, represented as $f(x) = \langle \kappa(x, :), f \rangle_{\mathcal{H}}$, where $\mathcal{H}$ is a Hilbert space, $f \in \mathcal{H}$, and there is an associated kernel $\kappa(x_i, x_j) = \langle \kappa(x_i, :), \kappa(:, x_j) \rangle_{\mathcal{H}}$. We briefly review functional gradients for scalar-output $f(x)$ (Scholkopf & Smola, 2002; Bagnell, 2012; Cheng et al., 2024), to set the stage for the vector outputs of interest here. Defining the evaluation functional $E_x(f) = \langle f, \kappa(:, x) \rangle_{\mathcal{H}} = f(x)$, the corresponding *functional gradient* is specified in terms of a scalar $\epsilon$ and $g \in \mathcal{H}$. Specifically, we have $E_x(f + \epsilon g) = E_x(f) + \epsilon E_x(g) = E_x(f) + \epsilon \langle \kappa(x, :), g \rangle_{\mathcal{H}} = E_x(f) + \epsilon \langle \nabla E_x, g \rangle_{\mathcal{H}} + O(\epsilon^2)$; the last equality effectively defines a functional gradient in terms of the linear term in a Taylor expansion of $E_x(f + \epsilon g)$ about $f$ (and here the $O(\epsilon^2)$ term is exactly zero). For $E_x$, the corresponding functional gradient in $\mathcal{H}$ is $\nabla E_x = \kappa(x, :)$, and $E_x$ is termed linear, as the second-order and higher terms are zero. A common nonlinear example used in kernel machines is $E_x(f) = \frac{1}{2}(y - f(x))^2$ for which via chain rule $\nabla E_x(f) = (f(x) - y)\kappa(x, :)$ (Scholkopf & Smola, 2002). For general $E(f)$, omitting the dependency on $x$ for notational simplicity, $\nabla E$ is termed the corresponding functional gradient.

Consider $E(f) = r(y, f(x))$ for general loss function $r(\cdot, \cdot)$. Again via chain rule, the corresponding functional gradient is $\nabla E = r'(y_i, f(x_i))\kappa(x_i, :)$, where $r'(y_i, f(x_i))$ is the usual derivative of $r(y_i, \gamma)$ wrt $\gamma$, evaluated at $\gamma = f(x_i)$. For data $\{(x_i, y_i)\}_{i=1,N}$, gradient descent in $\mathcal{H}$ is represented as $f_{k+1} = f_k - \frac{\alpha}{N} \sum_{j=1}^{N} r'(y_j, f_k(x_j))\kappa(x_j, :)$, where $\alpha$ is the learning step size. Consequently, $f_{k+1}(x) = \langle f_{k+1}, \kappa(:, x) \rangle_{\mathcal{H}} = f_k(x) - \frac{\alpha}{N} \sum_{j=1}^{N} r'(y_j, f_k(x_j))\kappa(x_j, x)$.

For $f(x)$ with $d'$-dimensional *vector* output, each component of which is modeled via the same type of RKHS setup as above, the derivative $r'(y_j, f(x_j))$ becomes a gradient wrt each component of $f(x_j)$. Considering $r(y_j, f(x_j)) = -\log p(Y = y_j | f(x_j))$, gradient descent becomes

$$f_{k+1}(x) = f_k(x) + \frac{\alpha}{N} \sum_{j=1}^{N} \nabla_f \log p(Y = y_j | f_k(x_j))\kappa(x_j, x), \tag{6}$$

where $\nabla_f \log p(Y = y_j | f_k(x_j))$ is a traditional gradient of $\log p(Y = y_j | f(x_j))$ wrt the components of $f(\cdot)$, evaluated at $f_k(x_j)$. This proves Lemma 1.

Moving to Proposition 1, the expression $\frac{1}{N} \sum_{j=1}^{N} \nabla_f \log p(Y = y_j | f_k(x_j))\kappa(x_j, x)$, which is the overall gradient responsible for the update of $f(x_j)$ in (6), may be interpreted as a *kernel-weighted average* (Hastie et al., 2009) of the $N$ isolated gradients $\nabla_f \log p(Y = y_j | f_k(x_j))$. In (6) the kernel average is implemented wrt RKHS kernels (positive semi-definite). Proposition 1 is a direct application of the Nadaraya-Watson modeling framework Nadaraya (1964); Watson (1964); Hastie et al. (2009), in which the kernel average is performed over the individual, sample-dependent gradients $\nabla_f \log p(Y = y_j | f_k(x_j))$, and the averaging is based on more-general representations $K_\lambda(x_i, x_j) / \sum_{j'=1}^{N} K_\lambda(x_i, x_{j'})$. The special case $K_\lambda(x_i, x_j) = \exp[\lambda(x_i^T x_j)]$ yields softmax attention.

### A.2  DETAILS ON THE GRADIENT UPDATES FOR REAL $Y$ AND GAUSSIAN $p(Y|f(x))$

We use the notation $f_{j,k} = f_k(x_j)$ to represent the function of interest after the $k$th step of gradient descent, evaluated at covariates $x_j$. For real-valued $Y$ and Gaussian $p(Y_i = y_i | f_i)$ (Ahn et al., 2024; von Oswald et al., 2023; Cheng et al., 2024; Ahn et al., 2023), we have

$$\nabla_{f_i} \log p(Y_i = y_i | f_i) = -\frac{1}{2\sigma^2} \nabla_{f_i} \|y_i - f_i\|^2 = \frac{1}{\sigma^2}(y_i - f_i), \tag{7}$$

and (1) yields

$$f_{j,k+1} = f_{j,k} + \underbrace{\frac{\alpha}{N} \sum_{i=1}^{N} (y_i - f_{i,k}) \kappa(x_i, x_j)}_{\Delta f_{j,k}} = f_{j,k} + \frac{\alpha}{N} \sum_{i=1}^{N} [y_i - \sum_{k'=0}^{k} \Delta f_{i,k'}] \kappa(x_i, x_j), \quad (8)$$

where we typically initialize as $f_{j,0} = 0_{d'}$, where $0_{d'}$ is a $d'$-dimensional vector of all zeros. Consequently, we also have $\Delta f_{i,0} = 0_{d'}$. The variance $\sigma^2$ is treated as a constant, and absorbed into the learning rate (effectively assuming that the variance $\sigma^2$ may be approximated as a constant for all contextual data of interest (von Oswald et al., 2023; Cheng et al., 2024)).

To make the connection to more-general $Y$ and $p(Y|f(x))$, we may re-express (8) as

$$\begin{aligned}
f_{j,k+1} &= f_{j,k} + \frac{\alpha}{N} \sum_{i=1}^{N} (y_i - \mathbb{E}(Y)_{|f_{i,k}}) \kappa(x_i, x_j) \\
&= f_{j,k} + \frac{\alpha}{N} \sum_{i=1}^{N} [y_i - \sum_{k'=0}^{k} \Delta_{\mathbb{E}_{i,k'}}] \kappa(x_i, x_j),
\end{aligned} \quad (9)$$

where $\Delta_{\mathbb{E}_{i,k'}} = \Delta f_{i,k'}$. As indicated in (8), $\Delta f_{i,k'}$ is the direct result of the attention mechanism.

As discussed above, real $Y$ and Gaussian $p(Y|f(x))$ is a special case, and in general $\mathbb{E}(Y)_{|f_{i,k}}$ is a *nonlinear* function of $f_{i,k}$. Moreover, for categorical $Y$, the expectation over outcomes $Y$, $\mathbb{E}(Y)_{|f_{i,k}}$, is replace by an expectation of learned embedding vectors (discussed next).

### A.3 DETAILS ON THE GRADIENT UPDATES FOR CATEGORICAL $Y$ AND SOFTMAX $p(Y|f(x))$

For categorical outcomes $Y$ and $p(Y = c|f(x)) = \exp[w_c^T f(x)] / \sum_{c'}^{C} \exp[w_{c'}^T f(x)]$,

$$\begin{aligned}
\nabla_{f_i} \log p(Y_i = y_i | f_i) &= \nabla_{f_i} \left[ f_i^T w_{y_i} - \log \sum_{c=1}^{C} \exp(f_i^T w_c) \right] \quad (10) \\
&= w_{y_i} - \frac{\sum_{c=1}^{C} w_c \exp(f_i^T w_c)}{\sum_{c'=1}^{C} \exp(f_i^T w_{c'})} \quad (11) \\
&= w_{y_i} - \sum_{c=1}^{C} p(Y_i = c | f(x_i)) w_c \quad (12) \\
&= w_{y_i} - \mathbb{E}(w_c)_{|f_i}, \quad (13)
\end{aligned}$$

and therefore in this case

$$f_{j,k+1} = f_{j,k} + \underbrace{\frac{\alpha}{N} \sum_{i=1}^{N} [w_{y_i} - \mathbb{E}(w_c)_{|f_{i,k}}] \kappa(x_i, x_j)}_{\Delta f_{j,k}} . \quad (14)$$

In general, the embedding vectors connected to the categories, $\{w_c\}_{c=1,C}$ are learned. Further, the expectation $\mathbb{E}(w_c)_{|f_{i,k}}$ is a *nonlinear* function of $f_{i,k}$, this necessitating approximations in the context of Transformer implementations of such a functional gradient descent. Stated differently, the Transformer *cannot* implement multiple steps of functional gradient descent for categorical $Y$, except for a one-layer model. These details are discussed in Sections A.5 and A.6

### A.4 DETAILS ON THE GRADIENT UPDATES FOR COUNT $Y$ AND POISSON $p(Y|f(x))$

The paper focuses on categorical $Y$, and seeks to make connections to real $Y$, to highlight relationships to prior work. Nevertheless, we briefly note one other special case, that underscores the generality of the approach. Specifically, for count-valued $Y$ and Poisson $p(Y_i = y_i | f_i) = \lambda_i^{y_i} \exp(-\lambda_i)/y_i!$, where $\lambda_i = \mathbb{E}(Y|X = x) = \exp(f(x_i))$ we can write

$$\nabla_{f_i} \log p(Y_i = y_i | f_i) = \nabla_{f_i} y_i f_i - \nabla_{f_i} \exp(f_i) = y_i - \exp(f_i), \tag{15}$$

and thus

$$f_{j,k+1} = f_{j,k} + \underbrace{\frac{\alpha}{N} \sum_{i=1}^{N} (y_i - \exp(f_{i,k})) \kappa(x_i, x_j)}_{\Delta f_{j,k}} = f_{j,k} + \frac{\alpha}{N} \sum_{i=1}^{N} [y_i - \mathbb{E}(y_i)_{|f_{i,k}}] \kappa(x_i, x_j), \tag{16}$$

where we made $\lambda_i = \exp(f_i)$ so $f_i \in \mathbb{R}$. Note that the formulation above readily extends to multivariate count-valued $Y$ by making the dimension of $f_i$ consistent to that of $Y$.

Concerning the above three subsections, note that for the Gaussian model for real $Y$, the update of $f_{j,k+1}$ may expressed in terms of a sum of prior $\{\Delta f_{j,k'}\}_{k'=0,k}$ (see (8), which as discussed below has important implications for (simplified) implementation. By contrast, for the categorical case of (3), the expression $\mathbb{E}(w_c)_{|f_{i,k}}$ is a *nonlinear* function of prior $\{\Delta f_{j,k'}\}_{k'=0,k-1}$, and similar issues hold for categorical $Y$. This adds significant complications, but as discussed below, it will also allow us to make explicit connections to all elements of the Transformer network (attention models *and* the feedforward elements (Vaswani et al., 2017)).

## A.5 Transformer Implementation for Real Observations

To make the connection to an attention-layer implementation for categorical $Y$ (Section A.6 below), we first design the attention layers for real $Y$. Further, we consider design of these attention layers for real $Y$ in a manner that aligns with how we will handle categorical $Y$. We then show why and how this can be simplified when considering real $Y$ and Gaussian $p(Y|f(x))$, thereby recovering the form used in prior work von Oswald et al. (2023); Cheng et al. (2024).

As the input to the first layer, consider the encoding $h_i^{(0)} = (x_i, 0_{d'}, y_i)^T$, where we initialize $f_i = 0_{d'}$, as done for the categorical case (and as done in von Oswald et al. (2023); Cheng et al. (2024) for real $Y$).

At each attention layer $W_Q$, $W_K$, $W_V$ and $P$ are designed such that $W_Q h_i^{(0)} = W_K h_i^{(0)} = (x_i, 0_{2d'})^T$ and $W_V h_i^{(0)} = (0_{d+d'}, \frac{\alpha}{N} y_i)^T$. These matrices are explicitly

$$W_Q = W_K = \begin{pmatrix} I_d & 0_{d \times d'} & 0_{d \times d'} \\ 0_{d' \times d} & 0_{d' \times d'} & 0_{d' \times d'} \\ 0_{d' \times d} & 0_{d' \times d'} & 0_{d' \times d'} \end{pmatrix}, \tag{17}$$

and

$$W_V = \begin{pmatrix} 0_{d \times d} & 0_{d \times d'} & 0_{d \times d'} \\ 0_{d' \times d} & 0_{d' \times d'} & 0_{d' \times d'} \\ 0_{d' \times d} & 0_{d' \times d'} & \frac{\alpha}{N} I_{d'} \end{pmatrix}. \tag{18}$$

The above design of these matrices yields attention layers that implement functional gradient descent for real $Y$.

For query $i$ and attention kernel $\kappa(x_i, x_j)$, attention yields $\sum_{j=1}^{N} W_V h_j^{(0)} \kappa(W_Q h_i^{(0)}, W_K h_j^{(0)}) = \frac{\alpha}{N} \sum_{j=1}^{N} y_j \kappa(x_i, x_j)$. Note that we provide this exposition for the first layer, but the same concept holds for all. In addition, we have used RKHS kernel attention, but the corresponding setup holds for softmax attention, from the perspective of Proposition 1.

We now define

$$P = \begin{pmatrix} 0_{d \times d} & 0_{d' \times d'} & 0_{d' \times d'} \\ 0_{d' \times d} & 0_{d' \times d'} & I_{d'} \\ 0_{d' \times d} & 0_{d' \times d'} & -I_{d'} \end{pmatrix}, \tag{19}$$

and with the skip connection we have

$$h_i^{(1)} = h_i^{(0)} + P \sum_{j=1}^{N} W_V h_j^{(0)} \kappa(W_Q h_i^{(0)}, W_K h_j^{(0)}) \tag{20}$$

$$= (x_i, 0_{d'}, y_i)^T + (0_d, \sum_{j=1}^{N} W_V h_j^{(0)} \kappa(W_Q h_i^{(0)}, W_K h_j^{(0)}), 0_{d'})^T$$

$$+ (0_{d+d'}, - \sum_{j=1}^{N} W_V h_j^{(0)} \kappa(W_Q h_i^{(0)}, W_K h_j^{(0)}))^T \tag{21}$$

$$= (x_i, \underbrace{\sum_{j=1}^{N} W_V h_j^{(0)} \kappa(W_Q h_i^{(0)}, W_K h_j^{(0)})}_{f_i^{(1)}}, \underbrace{y_i - \sum_{j=1}^{N} W_V h_j^{(0)} \kappa(W_Q h_i^{(0)}, W_K h_j^{(0)})}_{\nabla_{f_i} \sum_{j=1}^{N} \log p(Y=y_j | f_j^{(1)})})^T . \tag{22}$$

If we were interested in retaining $f_i^{(1)}$ for all inputs, this design would be appropriate. However, we are only interested in performing an estimate of $y_{N+1}$ for query $x_{N+1}$. Consequently, we only need $f_{N+1}$.

So motivated, the authors of von Oswald et al. (2023) had the insight of encoding the query $x_{N+1}$ by setting $y_{N+1} = 0_{d'}$ at the first input layer. Therefore, we observe that in (22) that with such an encoding the output of this attention layer for position $N+1$ is $(x_{N+1}, f_{N+1}^{(1)}, -f_{N+1}^{(1)})^T$. We see that in the case of real $Y$, when we are only interested in predictions for the query, it is redundant to encode $f_i$ and $\nabla_{f_i} \sum_{j=1}^{N} \log p(Y = y_j | f_j^{(1)})$. Consequently, the encoding is $h_i^{(0)} = (x_i, y_i)^T$ with $y_{N+1} = 0$. The matrices $W_Q$, $W_K$, $W_V$ and $P$ are reduced in size and modified accordingly (von Oswald et al., 2023). The output at the final attention layer at position $N+1$ is the *negative* of the desired $f_{N+1}$, the estimate of $y_{N+1}$ (von Oswald et al., 2023).

Concerning the above selections for $W_Q$, $W_K$, $W_V$ and $P$, these constitute only one such design, and many others are possible. For example, within attention kernels (and similarly for softmax attention), there are often inner products of the form $(h_i^{(\ell)})^T W_Q^T W_K h_j^{(\ell)}$. Any design of $W_Q$ and $W_K$ that yields the same $Q = W_Q^T W_K$ will manifest identical Transformer implementations. Similar issues hold wrt the design of $W_V$ and $P$ (von Oswald et al., 2023; Cheng et al., 2024).

### A.6 Details on Transformer implementation for categorical observations

**Single Layer of Attention, Steepest-Descent of Functional Gradient** For categorical $Y$ and softmax $p(Y|f(x))$, we first detail a single-layer model composed of attention alone, and demonstrate that it can exactly implement a step in the direction of the functional gradient.

For categorical observations, we encode at the input layer as $h_i^{(0)} = (x_i, 0_{d'}, w_{y_i} - \frac{1}{C} \sum_{c=1}^{C} w_c)^T$. We must maintain this form, even under the linear approximation of Section 4, because unlike for real $Y$, $f_i$ is *not* simply the negative of $\nabla_{f_i} \sum_{j=1}^{N} \log p(Y = y_j | f_j)$ (even if the outcome for $x_{N+1}$ is encoded to the zero vector). At the output of the Transformer the predicted $f_{N+1}$ is sent into the softmax operation over categories, and therefore we need to retain $f_{N+1}$.

For categorical observations, at each layer of the Transformer the matrices $W_Q$, $W_K$ and $W_V$ are the same as discussed in Section A.5 for real $Y$ with the same form of encoding. For ease of reading, they are repeated here:

$$W_Q = W_K = \begin{pmatrix} I_d & 0_{d \times d'} & 0_{d \times d'} \\ 0_{d' \times d} & 0_{d' \times d'} & 0_{d' \times d'} \\ 0_{d' \times d} & 0_{d' \times d'} & 0_{d' \times d'} \end{pmatrix}, \quad W_V = \begin{pmatrix} 0_{d \times d} & 0_{d \times d'} & 0_{d \times d'} \\ 0_{d' \times d} & 0_{d' \times d'} & 0_{d' \times d'} \\ 0_{d' \times d} & 0_{d' \times d'} & \frac{\alpha}{N} I_{d'} \end{pmatrix} . \tag{23}$$

When only considering a single attention layer, which exactly corresponds to one step of functional gradient descent, we use

$$P = \begin{pmatrix} 0_{d \times d} & 0_{d' \times d'} & 0_{d' \times d'} \\ 0_{d' \times d} & 0_{d' \times d'} & I_{d'} \\ 0_{d' \times d} & 0_{d' \times d'} & 0_{d' \times d'} \end{pmatrix} , \tag{24}$$

as in this case we only update $f_i^{(0)} \to f_i^{(1)}$, because this is all that is needed to feed into the softmax over categories at the output, to make predictions of categories (there is no need to update the corresponding functional gradients).

Concerning this single-layer implementation, from (5) recall that a first-order approximation of $\sum_{i=1}^{N} \log p(Y = y_i | f_i^{(0)} + g_\theta(i; \{h_j^{(0)}\}_{j=1,N}))$ indicates that the network $g_\theta(i; \{h_j^{(0)}\}_{j=1,N})$ should be aligned in the direction of the functional gradient $\nabla_{f_i} \sum_{j=1}^{N} \log p(Y = y_j | f_j^{(0)})$; in $p(Y = y_i | f_i^{(0)} + g_\theta(i; \{h_j^{(0)}\}_{j=1,N}))$ the additive $f_i^{(0)}$ represents the skip connection, and $g_\theta(i; \{h_j^{(0)}\}_{j=1,N})$ is implemented with an attention network.

Note, however, that the details of the functional gradient $\nabla_{f_i} \sum_{j=1}^{N} \log p(Y = y_j | f_j^{(0)})$ depend on the choice of kernel, if $f(x)$ is assumed to be within an RKHS family. More generally, softmax and other forms of attention may be considered from the perspective of Proposition 1 and Nadaraya-Watson kernel smoothing. We have found in practice, that nonlinear kernels like RBF typically yield results very similar to softmax attention (see Figure 1, for example). Therefore, from a practical perspective, the functional gradients appear to be consistent for different and sufficiently rich forms of nonlinear attention (assuming $f(x)$ is a nonlinear function).

**Multiple layers, categorical $Y$** For the multi-layered attention network, we seek to update the needed expectation iteratively, from layer to layer. Specifically, we express

$$\mathbb{E}(w_c)\big|_{f_i^{(\ell)}} = \sum_{\ell'=0}^{\ell} \Delta_{\mathbb{E}_i}^{(\ell')}, \quad \text{with } \Delta_{\mathbb{E}_i}^{(0)} = \frac{1}{C} \sum_{c=1}^{C} w_c. \tag{25}$$

Note that this is consistent with the update equation (8) associated with real $Y$, for which the latent function is updated incrementally with $\Delta f_{i,k}$, where there $k$ represents gradient descent step (which will link to Transformer layers). For real $Y$ and Gaussian $p(Y|f(x))$ with mean $f(x)$, the expectation (updated iteratively) is just $f(x)$. The iterative update of the expectation in (25) is the same idea, except that now $\mathbb{E}(w_c)\big|_{f_i^{(\ell)}}$ is *nonlinearly* related to $f_i^{(\ell)}$.

We consider two methods for the incremental update to the expectation:

- **Linear approximation**: In this approximation, we use

$$\Delta_{\mathbb{E}_i}^{(\ell)} \approx B^{(\ell)} \Delta f_i^{(\ell)}, \tag{26}$$

  where $\Delta f_i^{(\ell)}$ represents the incremental update to the latent function, manifested via attention. for categorical data, with a linear approximation to the update of the needed expectations over categories (Section 4), $W_Q$, $W_K$ and $W_V$ are unchanged from above at all layers, and

$$P^{(\ell)} = \begin{pmatrix} 0_{d \times d} & 0_{d' \times d'} & 0_{d' \times d'} \\ 0_{d' \times d} & 0_{d' \times d'} & I_{d'} \\ 0_{d' \times d} & 0_{d' \times d'} & -B^{(\ell)} \end{pmatrix}, \tag{27}$$

  where $B^{(\ell)} \in \mathbb{R}^{d' \times d'}$ matrix to be learned for each layer $\ell$. Comparing with (19) we see that this linear approximation is closely related to the attention design for real $Y$ (using the expanded encoding discussed in Section A.5). The difference is that for real $Y$ and Gaussian $p(Y|f(x))$ the incremental update to the latent expectation is exactly equal to the incremental update to the latent function. For categorical $Y$ and softmax $p(Y|f(x))$, under linearization, we must additionally learn the matrix $B^{(\ell)}$ at each layer.

- **Nonlinear representation with feedforward element** In this setup, each block of the Transformer is composed of an attention network, exactly the same as that used for the single-layer setup introduced above. Specifically, $W_Q$, $W_K$ and $W_V$ remain unchanged, and for $P$ we use (24). The input at position $i$ into layer $\ell + 1$ is

$$\text{Input to layer } \ell + 1: \quad h_i^{(\ell)} = \left( x_i, \sum_{\ell'=0}^{\ell} \Delta f_i^{(\ell')}, w_{y_i} - \sum_{\ell'=0}^{\ell} \Delta_{\mathbb{E}_i}^{(\ell')} \right)^T. \tag{28}$$

The output of the attention layer at position $i$ is $(0_{d'}, \Delta f_i^{(\ell+1)}, 0_{d'})^T$; when this is added to the input $h_i^{(\ell)}$ as a result of the skip connection, we have

$$\text{output of attention + skip connection}: \left( x_i, \sum_{\ell'=0}^{\ell+1} \Delta f_i^{(\ell')}, w_{y_i} - \sum_{\ell'=0}^{\ell} \Delta_{\mathbb{E}_i}^{(\ell')} \right)^T . \qquad (29)$$

We now wish to add $\Delta_{E_i}^{(\ell+1)}$ to the expectation, based on our updated form of the latent function $f_i^{(\ell+1)} = \sum_{\ell'=0}^{\ell+1} \Delta f_i^{(\ell')}$. Rather than using the above linear approximation, we consider a feedforward element with parameters $\phi$:

$$\Delta_{E_i}^{(\ell+1)} = \text{FF}_\phi \left( \sum_{\ell'=0}^{\ell+1} \Delta f_i^{(\ell')}, w_{y_i} - \sum_{\ell'=0}^{\ell} \Delta_{\mathbb{E}_i}^{(\ell')} \right) , \qquad (30)$$

where $\phi$ denote the parameters of the feedforward (FF) element, which is the same for all positions $i$ at a given layer. Concerning (30), we use the updated representation of $f_i$ and the prior representation of the gradient, $w_{y_i} - \sum_{\ell'=0}^{\ell} \Delta_{\mathbb{E}_i}^{(\ell')}$, as inputs to a nonlinear FF element, and we use this to estimate $\Delta_{E_i}^{(\ell+1)}$. Utilizing the skip connection associated with the FF element, the representation for $\Delta_{E_i}^{(\ell+1)}$ can be placed in the proper position. Consequently, ideally, one Transformer block (attention plus FF, with skip connections associated with each), will yield the output of block $\ell + 1$:

$$\text{output of GD Transformer block } \ell + 1 : \left( x_i, \sum_{\ell'=0}^{\ell+1} \Delta f_i^{(\ell')}, w_{y_i} - \sum_{\ell'=0}^{(\ell+1)} \Delta_{\mathbb{E}_i}^{(\ell')} \right)^T . \qquad (31)$$

## A.7 DETAILS ON TRAINED TF DESIGNS

For the Trained TF variants of the above GD-motivated setups, the model is free to learn parameters, without restrictions. Specifically, the attention matrices $W_Q$, $W_K$, $W_V$ and $P$ are learned without restrictions. Additionally, there is the matrix $Z$ at the output, the results of which then feed into the softmax model at the output; $Z$ is also learned. When they are *not* set as one-hot vectors, the category-dependent vectors $w_c$ are also learned.

The Trained TF version of the linearized multi-layered model above simple corresponds to multiple layers of attention, with all parameters learned. While with GD we restrict the form of $P^{(\ell)}$ as in (27), for Trained TF, the attention matrices have complete freedom wrt the parameters learned.

The Trained TF version of the nonlinear representation corresponds to the original design of the Transformer (Vaswani et al., 2017), in which the attention layer is followed by FF elements, with skip connections employed with both. The only element missing from the original Transformer is layer normalization, which we avoid because it is not evident in the GD-based formulation.

The Trained TF, with attention layers alone (corresponding to the aforementioned linearized multi-layered GD) or with blocks composed of an attention layer and FF element (corresponding to the aforementioned multi-layered GD with FF elements) offer significant modeling power. However, the training of these models via Adam (Kingma & Ba, 2015) was found to be challenging from a random parameter initialization. As discussed in the main paper, parameter initialization was found to be very important for Trained TF, consistent with prior work (Liu et al., 2023). In particular, to get good results with Trained TF, we found initialization with the appropriate GD variant is important.

## A.8 ADDITIONAL IMAGENET RESULTS

We present in Figure 5 results for the same Transformer considered in the main body of the paper, which was trained using context sizes $N = 50$. Here it is tested using context blocks of size $N = 15$, with 5 image classes per contextual block, and 3 images selected (uniformly) at random per context set (no changes were made to the Transformer after training, despite the fact that they are applied now to much small contextual sets). Results are shown with softmax attention (left in Figure 5) and with linear attention (right in Figure 5). Because of the high quality of the VGG features representing

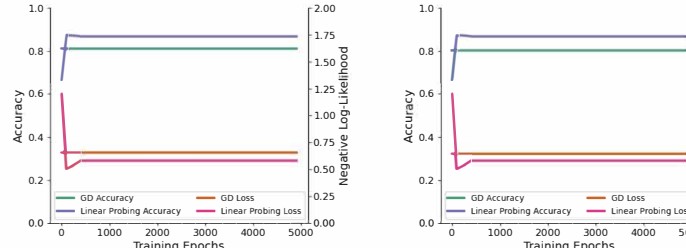

Figure 5: Results for the ImageNet dataset, considering $N = 15$ for the test contextual datasets (the Transformer was trained with $N = 50$, and therefore there is a mismatch between the training and test context size $N$ for the Transformer. Since the Transformer was trained with $N = 50$, the Transformer (GD) results are constant for all of what are here shown as training epochs, which are for the linear probing comparison model. Left: softmax attention; Right: linear attention.

the covariates $x_i$, linear attention works well here. However, as discussed in the main paper, for the linear attention results the factor $1/N$ was rescaled within the software at test time, to account for the change in the context size (from that used when training). In contrast, for softmax attention no changes were made to the trained model (discussed in the main body of the paper).

In Figure 5 we observe that the Transformer (here based on GD, for one layer) performs well for both linear attention and softmax attention. While the results do not reach those of linear probing, they are relatively close. Unlike linear probing, the Transformer is performing few-shot learning with no parameter refinement when testing (except for the $1/N$ for linear attention). Further, the Transformer was trained using different covariate statistics than that seen at testing (recall that it was trained using images from 900 of the ImageNet classes, and applied here to the remaining 100 image classes). In contrast, linear probing trains on the test data, anew for each contextual block.

