# OpenReview forum: "Functional Gradients and Generalizations for Transformer In-Context Learning"
_ICLR.cc/2025/Conference — Submitted to ICLR 2025_

### Official Review · Reviewer_SD88 · 2024-10-27

**Soundness:** 2
**Presentation:** 1
**Contribution:** 2
**Rating:** 3
**Confidence:** 3

**Summary:**

This paper follows a line of work that seeks to understand the emergence of in-context learning ability in transformers through their ability to learn to solve problems in which the context consists of $N$ input and label pairs and an $N+1$-th query input, which are assumed to all be drawn from the same distribution, and the goal is to predict the missing label of the query input. Different from most prior works that consider labels that are linear functions of the inputs, or functions from an RKHS, this work extends to the case where $y_i \sim p(Y|f(x_i))$, where the latent function $f$ may not belong to an RKHS. The paper performs analysis on the expressivity of shallow transformers to solve such tasks and provides supporting experiments.

**Strengths:**

- The paper addresses an important topic, and its motivation to extend prior works to more general scenarios, with random labels, general latent $f$, and nonlinear attention, is strong.

- The analytic framework presented is promising, and the results can likely be impactful if better formulated.

**Weaknesses:**

- The writing is too dense, making it very difficult to parse. There is too much math in the problem formulation and analysis. The analysis should be more often stated in formal results and the surrounding text used to discuss intuition. Sections should be more often broken into subsections with each subsection playing a clear and distinct role, especially in the experiments.

- Unless I am missing something, Proposition 1 does not show that one layer of attention expresses functional gradient descent in any sense, contrary to earlier claimed contributions. The result concerns only writing the NW average gradient, and is unrelated to attention, from my understanding. As a result, the theoretical contributions are underwhelming.

- The conclusions from the experiments are not clear.

- Minor:
       The usage of real vs categorical is somewhat confusing because categorical one-hot encodings are real;
       typos in lines 53, 117, 271

**Questions:**

n/a

---

### Official Review · Reviewer_76FB · 2024-11-03

**Soundness:** 2
**Presentation:** 2
**Contribution:** 1
**Rating:** 3
**Confidence:** 4

**Summary:**

This paper analyzes Transformer-based in-context learning through the lens of functional gradient descent, showing that a single attention layer can implement gradient steps for both regression and classification tasks, with a particular focus on extending previous work to handle categorical outputs using softmax probability models.

**Strengths:**

The paper provides a clear mathematical connection between functional gradients and transformer attention mechanisms, particularly for categorical data.

**Weaknesses:**

1. I don't think it's meaningful to design the transformer architecture for a specific task. And your experiments are all on shallow transformers and very limited tasks, which are insufficient to show that the proposed transformer architectures are better than the mainstream structures.

2. There's not much theoretical insight. Their main theoretical results are relatively straightforward applications of RKHS/Nadaraya-Watson Averaged gradient descent, which has been well-studied in previous work.

**Questions:**

If each layer is performing one step of GD, why 2-layer transformers can't outperform 1-layers ones? Did you try more layers and harder tasks? It's clearly impossible that GD converges in one step for all tasks.

**Details Of Ethics Concerns:**

No ethics concerns.

---

### Official Review · Reviewer_Frxb · 2024-11-05

**Soundness:** 2
**Presentation:** 1
**Contribution:** 2
**Rating:** 5
**Confidence:** 2

**Summary:**

This paper develops a theoretical framework to analyze transformers’ ability to perform in-context learning (ICL) through functional gradient descent, specifically focusing on scenarios involving categorical data. The authors propose that a single layer of attention in transformers can approximate a gradient step, aligning with functional gradient descent under a kernel-weighted averaging scheme. They extend this insight to categorical outputs modeled through softmax probabilities, connecting transformer operations with the Nadaraya-Watson estimator. The work also explores multi-layered extensions, concluding that one layer often suffices for effective performance in in-context learning tasks, supported by synthetic and ImageNet experiments.

**Strengths:**

The paper addresses an essential topic in the theoretical analysis of transformers and ICL, filling a gap often left by experimental studies.

**Weaknesses:**

1. The paper’s organization and presentation are challenging to follow. Numerous notations are introduced prematurely, even in the abstract and introduction, without sufficient explanation until later sections.

2. The main conclusions of the paper are unclear. Although the authors provide analyses from various perspectives, there is no cohesive summary or highlighted takeaways connecting the derivations to significant findings.

3. The system setup is ambiguous. It is unclear whether the setup in Sections 2, 3, and 4 remains consistent throughout or varies between sections.

4. Sections 3 and 4 lack propositions or theorems to underscore main conclusions, making it difficult to discern the primary results in these sections.

**Questions:**

Please refer to the weaknesses listed above.

---

### Official Review · Reviewer_86SG · 2024-11-06

**Soundness:** 2
**Presentation:** 2
**Contribution:** 1
**Rating:** 3
**Confidence:** 3

**Summary:**

The paper analyses in-context learning (ICL) in Transformers as functional gradient descent steps. These functional gradient steps are first analyzed for functions belonging to a reproducing kernel Hilbert space (RKHS) and subsequently generalized as Nadaraya-Watson kernel weighted averages. This allows for the analysis of softmax attention for ICL. The authors use this to design a single-layer softmax attention layer to implement one step of Nadaraya-Watson averaged gradient descent. Brief experiments show that a single-layer model is effective for ICL, even in real-life data.

**Strengths:**

The paper generalizes the idea of [1] beyond functions from an RKHS by linking it to the Nadaraya-Watson kernel estimate. Despite pointing this out a strength, I have mixed opinions on this because this idea seems natural and has also been explored and different explanations have been provided in prior works (e.g., [2]).

[1] Cheng, Xiang, Yuxin Chen, and Suvrit Sra. "Transformers implement functional gradient descent to learn non-linear functions in context." arXiv preprint arXiv:2312.06528 (2023).

[2] Collins, Liam, et al. "In-context learning with transformers: Softmax attention adapts to function lipschitzness." arXiv preprint arXiv:2402.11639 (2024).

**Weaknesses:**

The technical contributions of the paper are limited. Lemma 1 seems true simply by definition, thus, the theoretical contributions seem insufficient. Further, the statement and implications of Proposition 1 are unclear. Finally, the model design is also unclear and, at first glance, appears to have no trainable parameters in the attention layer.

Next, the paper seems to claim that single-layer transformer suffices for in-context learning, and multiple-layers are unnecessary. This is neither theoretically justified nor are there sufficient experiments to justify this strong claim.

Finally, In the experiments on real-life data, the outputs of the VGG classifier are fed as features into the (single-layer) transformer. It is unclear if most of the “actual classification” is done by the VGG network, in which case the strong results are primarily due to the VGG network and not due to in-context learning.

**Questions:**

1. In proposition 1, what space does $f$ lie in? Is the derivative well-defined in this space?

2. Is the implication of Proposition 1 that the given statement is the estimate of the gradient? If not, what is the claim?

3. What are the trainable parameters in the single-layer model?

4. Do single-layer transformers work sufficiently well for ICL if the input features are not as learned, i.e., if one gives the output of the middle layers of the VGG network instead? If not, could the authors provide stronger justification for their conclusion that a single attention layer is sufficient?

---

### Official Review · Reviewer_fB5R · 2024-11-07

**Soundness:** 2
**Presentation:** 2
**Contribution:** 1
**Rating:** 3
**Confidence:** 3

**Summary:**

The paper studies in-context learning (ICL) of transformers from the perspective of functional gradient descent (FGD), where observables $Y$ are generated from covariates $x$ according to a distribution $P(Y|f(x))$ and $f$ is a latent mapping. It is shown that the Nadaraya-Watson kernel-weighted average (generalizing FGD in the RKHS setting) can be exactly implemented using an attention layer; in particular, softmax attention suffices when $Y$ is categorical and $P$ is a generalized linear model. Approximate multi-layer contructions are also given. Moreover, experiments show that a single-layer transformer is an effective in-context learner for synthetic and image classification tasks.

**Strengths:**

The functional gradient descent viewpoint is a very general framework encompassing multiple previous works which show e.g. that the forward pass of depth $L$ transformers can perform $L$ steps of (ordinary or preconditioned) gradient descent on Gaussian data. The connection to FGD in RKHS is intuitive, and it is interesting to see that this insight yields a similar construction for categorical data. A range of synthetic and real-world experiments are given to study the ICL efficacy of one- and two-layer transformers.

**Weaknesses:**

There have been numerous papers claiming that transformers (can or do) perform some sort of optimization algorithm in context during their forward pass. Such results are nice as known guarantees of the algorithms can be immediately transferred over to ICL. However, without theoretical or empirical evidence showing that such behavior indeed manifests in trained transformers --  for example, by proving optimization guarantees for the given constructions or experimentally demonstrating behavior indicative of such algorithms -- the paper cannot serve as a convincing contribution to our understanding of ICL, rather only confirming the expressivity of the transformer architecture. More specific weaknesses are detailed below.

* No optimization analysis or loss landscape analysis is given, so it is unclear whether the proposed parameter constructions can actually arise in trained transformers. Some existing works have already proved convergence of pretraining dynamics for one layer of linear attention with linear [1] or nonlinear mappings [2] as well as one layer of softmax attention [3,4], so it is not unreasonable to consider such analyses at least for the one-layer categorical case.
* The equivalence is only exact for one step of FGD with a single attention layer, except for the real/Gaussian case which was established in previous works. While approximate extensions to multi-layer attention are given in Section 4, no analysis of the approximation error is given, so that whether multi-layer FGD is indeed a valid model of the forward pass of a deep transformer is left unclear.
* While the experiments seems to indicate some similarities between GD and fully trained transformers, the connection is never fully justified. For example, it is claimed that "We attribute (ii) to the fact that the attention layer and skip connection manifest an effective functional gradient step," however the efficacy of the one-layer model does not tell us anything about what algorithm the layer is actually implementing. Even if this is true, the theoretical analysis does not explain why one step of FGD is sufficient for good performance (compared to more steps). Moreover, the observation that the trained model does not move from the GD initialization indicates that GD is a stationary point, but does not imply that training transformers from scratch converges to GD.
* The writing of the paper needs to be improved. For example, it is unclear what Section 4 is achieving, as the constructions are omitted and there are no precise theoretical guarantees. Also, a lot of text throughout the body of the paper is spent on comparing to existing works on the real/Gaussian setting, which could be more concisely explained in a Related Works section or a table.
* The FGD perspective for ICL has already been proposed in [5], which also considers multi-head attention and performs loss landscape analysis. While the paper mentions that the distribution for $Y$ can be more general, I would expect a much more detailed comparison to [5] in order to be sufficiently convinced of the novelty of this paper.

[1] Zhang et al. Trained Transformers Learn Linear Models In-Context. 2023.

[2] Kim et al. Transformers Learn Nonlinear Features In Context: Nonconvex Mean-field Dynamics on the Attention Landscape. ICML 2024.

[3] Huang et al. In-Context Convergence of Transformers. 2023.

[4] Chen et al. Training Dynamics of Multi-Head Softmax Attention for In-Context Learning: Emergence, Convergence, and Optimality. COLT 2024.

[5] Cheng et al. Transformers Implement Functional Gradient Descent to Learn Nonlinear Functions in Context. ICML 2024.

**Questions:**

Please see Weaknesses.

---

### Official Review · Reviewer_cyD7 · 2024-11-08

**Soundness:** 2
**Presentation:** 1
**Contribution:** 1
**Rating:** 3
**Confidence:** 4

**Summary:**

This paper considers the problem of in-context learning from the perspective of functional gradient descent. The authors draw a line between the Nadaraya-Watson average and the softmax attention. The authors also generalize the construction of the one-step functional gradient descent layer from the regression case to the classification case under the multinomial logistic regression model. The authors also conduct numerical experiments to verify that Transformers are doing functional gradient descent under some initialization. They also find out that the performance of a single-layer Transformer is good enough under some circumstances.

**Strengths:**

1. This paper considers a softmax (or multinomial logistic regression) model that naturally generalizes the previous conclusions for linear/kernelized regression.
2. Numerical experiments verify that under some initializations, the trained Transformer behaves coincidently with the constructed Transformer of the functional gradient descent.

**Weaknesses:**

1. The presentation is not clear. As a paper based on certain constructions of Transformers, it does not specify the Transformer model structure and the parameters at the beginning. Also, the main conclusion that the Transformer can implement GD is not strictly formalized in a theorem or proposition, while two other trivial conclusions are stated as theorems/propositions (Lemma 1 and Proposition 1).
2. The contribution is overclaimed and is marginal compared to existing works. The authors consider the classification case (and some other cases not included in the main text), but a) the problem does not provide further insights to understand the dynamics of ICL compared to previous works; b) the contribution is marginal compared to that in Cheng et al. (2023) since, for example, the classification is done via multinomial logistic regression. See the Questions Part 1.
3. The conclusion is not solid enough. There are two types of theoretical research drawing the link between the gradient descent and the in-context learning of Transformers: the first kind is that \textit{some} constructed Transformers \textit{can} adopt GD to do ICL, and the second is that \textit{trained} Transformers (under some constraints) \textit{will} adopt GD to do ICL. This paper is doing the first kind of theory, which itself doesn't necessarily become a problem as long as it is sufficiently verified by numerical experiments. However, this paper doesn't include enough experiments (for example, directly examining the model weights). See the Questions Part 2.

References: X. Cheng, Y. Chen, and S. Sra. Transformers implement functional gradient descent to learn nonlinear functions in context.

**Questions:**

1. From my point of view, the reason that we are proving Transformers can accomplish simple tasks such as linear regression in the 2020s is only because those simple function classes may provide some insights into understanding the dynamics of Transformer. The motivation to study the multi-classification problem is not fully characterized in the paper. What's the difference between the multi-classification and the regression problem in your ICL study (especially when there have already been cases studying the functional space such as Cheng et al. 2023)?
2. Current arguments are not strong enough (see the Weaknesses Part 3). Can you provide any further evidence supporting the line between the GD and the ICL? a) The loss landscape analysis to prove that the trained Transformer does converge to certain points like that in Zhang et al. 2023, Ahn et al. 2023, and Wu et al. 2023. b) Numerical experiments to examine the coincidence between the constructed Transformer and the trained Transformer like Figure 2 in Ahn et al. 2023.

References: K. Ahn, X. Cheng, H. Daneshmand, and S. Sra. Transformers learn to implement preconditioned gradient descent for in-context learning.

R. Zhang, S. Frei, and P.L. Bartlett. Trained transformers learn linear models in-context.

J. Wu, D. Zou, Z. Chen, V. Braverman, Q. Gu,  and P.L. Bartlett. How many pretraining tasks are needed for in-context learning of linear regression?

---

### Official Review · Reviewer_5bAC · 2024-11-12

**Soundness:** 3
**Presentation:** 2
**Contribution:** 2
**Rating:** 5
**Confidence:** 3

**Summary:**

The paper explores the role of functional gradient descent in enhancing Transformers for in-context learning tasks. It builds upon the RKHS (Reproducing Kernel Hilbert Space) framework to offer new perspectives on kernel-weighted averaging and softmax attention.

It demonstrates that a Transformer's forward pass can be viewed as kernel-averaged functional gradient descent. The study extends functional gradient frameworks to softmax attention and non-RKHS latent functions. Furthermore, it highlights the effectiveness of single-layer Transformer models for categorical data, validated through experiments on both synthetic and real-world datasets.

**Strengths:**

- The study generalizes existing theories to non-RKHS settings, broadening its applicability.

-  The paper includes rigorous experiments, including applications to ImageNet, showcasing the practical utility of the proposed methods.

- The mathematical derivations are clearly presented.

**Weaknesses:**

In my opinion, the technical novelty is limited in the following aspects:

-  The methodology appears to refine or slightly extend known results (e.g., functional gradient descent for in-context learning in RKHS).
- The proof of Lemma 1 is not particularly novel; it builds on well-known principles from RKHS and functional gradient descent. Similar derivations can be found in classical texts on kernel methods (e.g., Schölkopf \& Smola, 2002).
- The derivation of the Transformer implementation in Sections 3 and 4 closely aligns with prior works such as von Oswald et al. (2023), Cheng et al. (2024), and Ahn et al. (2023).

**Questions:**

Q1. How does this work provide fundamentally new theoretical or empirical insights beyond prior studies such as von Oswald et al. (2023) or Cheng et al. (2024)? What are the specific, measurable benefits of your generalizations in real-world applications?

Q2. In Lemma 1, should the step size be $\alpha$ instead of $\alpha/N$?

Q3.  The derivation in Lemma 1 closely mirrors foundational RKHS results. As such, it may not add substantial theoretical novelty. Can you comment on this?

Q4.  In Section 3, the analysis heavily relies on the assumption that the first-order term dominates, while the higher-order residual $\delta$ is small enough to be negligible. Does this assumption hold in highly non-linear settings or deeper Transformer architectures?

Q5. In Section 4, the design relies on weight matrices and kernelized attention to approximate functional gradients. However, softmax normalization can sometimes amplify small variations in input values. How does the model ensure numerical stability in these cases?

Q6.  In Section 4, an additive linear approximation
$\Delta^{(\ell)}_{E_i}$
    is introduced to model the expectation updates. How does the linear approximation affect convergence and the accuracy of the learned function, particularly when the underlying data distributions or transformations are highly non-linear?

Q7. In Section 5, how does the model's performance change when the contextual data blocks and query set distributions significantly differ?

---

### Meta-Review · Area_Chair_4RMv · 2024-12-20

**Metareview:**

This paper examines the problem of in-context learning from the perspective of functional gradient descent. The authors show that a single layer of attention in transformers can be designed to implement functional gradient descent, extending prior works. The authors also conduct numerical experiments to verify the analysis in the paper.

The reviews for the paper were mostly negative with primary concerns being: (1) limited technical novelty (2) unclear presentation and (3) unclear empirical analysis. I agree with the reviewers that the novelty of the paper is somewhat limited and the presentation needs major improvement. Unfortunately, the authors did not respond to the feedback. I recommend rejection.

**Additional Comments On Reviewer Discussion:**

The authors chose not to respond to reviewer's feedback.

---

### Decision · Program_Chairs · 2025-01-22

Reject